# GenerSpeech: Towards Style Transfer for Generalizable Out-Of-Domain Text-to-Speech

**Rongjie Huang**[*]
Zhejiang University
rongjiehuang@zju.edu.cn

**Yi Ren**[*]
Sea AI Lab
renyi@sea.com

**Jinglin Liu**
Zhejiang University
jinglinliu@zju.edu.cn

**Chenye Cui**
Zhejiang University
chenyecui@zju.edu.cn

**Zhou Zhao**[†]
Zhejiang University
zhaozhou@zju.edu.cn

## Abstract

Style transfer for out-of-domain (OOD) speech synthesis aims to generate speech samples with unseen style (e.g., speaker identity, emotion, and prosody) derived from an acoustic reference, while facing the following challenges: 1) The highly dynamic style features in expressive voice are difficult to model and transfer; and 2) the TTS models should be robust enough to handle diverse OOD conditions that differ from the source data. This paper proposes GenerSpeech, a text-to-speech model towards high-fidelity zero-shot style transfer of OOD custom voice. GenerSpeech decomposes the speech variation into the style-agnostic and style-specific parts by introducing two components: 1) a multi-level style adaptor to efficiently model a large range of style conditions, including global speaker and emotion characteristics, and the local (utterance, phoneme, and word-level) fine-grained prosodic representations; and 2) a generalizable content adaptor with Mix-Style Layer Normalization to eliminate style information in the linguistic content representation and thus improve model generalization. Our evaluations on zero-shot style transfer demonstrate that GenerSpeech surpasses the state-of-the-art models in terms of audio quality and style similarity. The extension studies to adaptive style transfer further show that GenerSpeech performs robustly in the few-shot data setting. [3]

## 1  Introduction

Text-to-speech [39, 37, 20, 13] aims to generate almost human-like audios using text, which attracts broad interest in the machine learning community. These TTS models have been extended to more complex scenarios, including multiple speakers, emotions, and styles for expressive and diverse voice generation [14, 30, 4]. A growing number of applications [51, 11, 50, 52], such as voice assistant services and long-form reading, have been actively developed and deployed to real-world speech platforms.

Increasing demand for personalized speech generation challenges TTS models especially in unseen scenarios regarding domain shifts. Unlike typically controllable speech synthesis, style transfer for generalizable out-of-domain (OOD) text-to-speech aims to generate high-quality speech samples with unseen styles (e.g., timbre, emotion, and prosody) derived from an acoustic reference (i.e.,

---

[*]Equal contribution.

[†]Corresponding author

[3]Audio samples are available at `https://GenerSpeech.github.io/`

36th Conference on Neural Information Processing Systems (NeurIPS 2022).

custom voice), which is hampered by two major challenges: 1) style modeling and transferring: the high dynamic range in expressive voice is difficult to control and transfer. Many TTS models only learn an averaged distribution over input data and lack the ability to fine-grained control style in speech sample; and 2) model generalization: when the distributions of style attributes in custom voice differ from training data, the quality and similarity of synthesized speech often deteriorate due to distribution gaps. While current TTS literature [48, 44, 27, 6] has considered improving model capability towards OOD text-to-speech, they fail to address the above challenges fully. Specifically,

- **Style modeling and transferring**. Researchers propose several methods to model and transfer the style attributes: 1) Global style token. Frequently mentioned style transfer work [48] introduces the idea of global style token (GST) and its derivative [44] further manages to memorize and reproduce the global scale style features of speech. 2) Fine-grained latent variables. Sun et al. [42, 41] adopt VAE to represent the fine-grained prosody variable, naturally enabling sampling of different prosody features for each phoneme. Li et al. [27] utilizes both the global utterance-level and the local quasi-phoneme-level style features of the target speech. However, these methods are pretty limited in capturing differential style characteristics and fail to simultaneously reflect the correct speaker identity, emotion, and prosody ranges.

- **Model generalization**. Researchers investigate paths to improve model generalization towards OOD custom voice: 1) Data-driven method. A popular approach [16, 34] to improve the robustness of the TTS model is to pre-train on a larger dataset consisting of various speeches to expand data distribution. Unfortunately, this data-hungry approach requires many audio samples and corresponding transcripts, which is often costly or even impossible. 2) Style adaptation. Chen et al. [6] adapts new voice by finetuning on the limited adaptation data with diverse acoustic conditions. Several works [31, 15] adopt meta-learning to adapt to new speakers that have not been seen during training. However, style adaptation relies on a strong assumption that the target voice is accessible for model adaptation, which does not always hold in practice. How to generalize for zero-shot out-of-domain speech synthesis is still an open problem.

An intuitive way [25, 29] to achieve better generalization is to decompose a model into the domain-agnostic and domain-specific parts via disentangled representation learning. To address the above challenges in style transfer of OOD custom voice, we propose **GenerSpeech**, a **gener**alizable text-to-**speech** model for high-fidelity zero-shot style transfer of out-of-domain voice, including several techniques to model and control the style-agnostic (linguistic content) and style-specific (speaker identity, emotion, and prosody) variations in speech separately: 1) **Multi-level style adaptor**. We propose the multi-level style adaptor for the global and local stylization of the custom utterance. Specifically, a downstream wav2vec 2.0 encoder generates the global latent representations to control the speaker and emotion characteristics. Furthermore, three differential local style encoders model the fine-grained frame, phoneme, and word-level prosodic representations without explicit labels. 2) **Generalizable content adaptor**. We propose the mix-style layer normalization (MSLN) to effectively eliminate the style attributes in the linguistic content representation and predict the style-agnostic variation, to improve the generalization of GenerSpeech.

We conduct experiments on zero-shot style transfer for out-of-domain text-to-speech synthesis in the OOD testing sets. Experimental results demonstrate that GenerSpeech achieves new state-of-the-art zero-shot style transfer results for OOD text-to-speech synthesis. Both subjective and objective evaluation metrics show that GenerSpeech exhibits superior audio quality and similarity compared with baseline models. The extension studies to adaptive style transfer further prove that GenerSpeech performs robustly in the few-shot data setting.

## 2 Related Works

### 2.1 Style Modeling and Transferring in Text-to-Speech

Style modeling and transferring have been studied for decades in the TTS community: The idea of global style tokens [48] represents a success in controlling and transferring the global style. Sun et al. [41, 42] further study a way to include a hierarchical, fine-grained prosody representation. Li et al. [27] additionally adopt a multi-scale reference encoder to explore the phoneme-level style modeling. The models mentioned above resort to the autoregressive generation of mel-spectrogram, suffering from slow inference speed and a lack of robustness.

Recently, non-autoregressive TTS models [23, 12, 18, 17] have been proposed and significantly speed up mel-spectrogram generation, and researchers have experimented on style modeling and transferring in parallel: Meta-StyleSpeech [31] generally adopts a speech encoding network for multi-speaker TTS synthesis. SC-GlowTTS [3] proposes a speaker-conditional architecture that explores a flow-based decoder in a zero-shot scenario. Styler [24] models style factor via decomposition. However, these methods focus on a limited area of style modeling, without simultaneously considering the speaker identity, emotion, and prosody variation.

## 2.2 Domain Generalization

Previous works on text-to-speech adaptation [6, 31] rely on a strong assumption that the target voices are accessible for model adaptation, which does not always hold in practice. In many scenarios, target custom voice is difficult to obtain or even unknown before deploying the TTS model, which is considered a more challenging problem: domain generalization (DG).

Many zero-shot DG methods are based on the idea of aligning features between different sources, with the hope that the model can be invariant to domain shift given unseen data. Li et al. [28] resort to adversarial learning with auxiliary domain classifiers to learn features that are domain-agnostic. Li et al. [26] use meta-learning to simulate train/test domain shift during training and jointly optimize the simulated training and testing domains. Ulyanov et al. [43] add Instance normalization layers to eliminate instance-specific style discrepancy in the field of image style transfer. However, all these methods focus on the image domain. In contrast, our work focuses on style generalization in text-to-speech synthesis, which is relatively overlooked.

# 3   GenerSpeech

In this section, we first define and formulate the generalizable text-to-speech model for zero-shot style transfer of out-of-domain custom voice. We then overview the proposed GenerSpeech, following which we introduce several critical components including the generalizable content adaptor and multi-level style adaptor. Finally, we present the pre-training, training and inference pipeline of GenerSpeech for high-fidelity out-of-domain text-to-speech synthesis.

## 3.1   Problem formulation

Style transfer of out-of-domain (OOD) custom voice aims to generate high-quality and similarity speech samples with unseen style (e.g., speaker identity, emotion, and prosody) derived from a reference utterance, which has different acoustic conditions from training data.

## 3.2   Overview

We adopt one of the most popular non-autoregressive TTS models FastSpeech 2 [37] as the model backbone. The overall architecture of GenerSpeech has been presented in Figure 6. An intuitive way [25, 29] to achieve better generalization is to decompose a model into the domain-agnostic and domain-specific parts via disentangled representation learning. Therefore, to improve generalization in text-to-speech synthesis, we design several techniques to model the style-agnostic (linguistic content) and style-specific (e.g., speaker identity, emotion, and prosody) variations in speech separately:

1) to improve model generalization, we propose mix-style layer normalization(MSLN) to eliminate the style information in the linguistic content representation. 2) to enhance modeling and transferring style attributes, we introduce a multi-level style adaptor consisting of a global encoder for speaker and emotion feature embeddings and three differential (frame, phoneme, and word-level) local encoders for prosodic style representations. 3) to reconstruct details in these expressive speech samples, we include a flow-based post-net [38] to refine the transform decoder output and generate fine-grained mel-spectrograms.

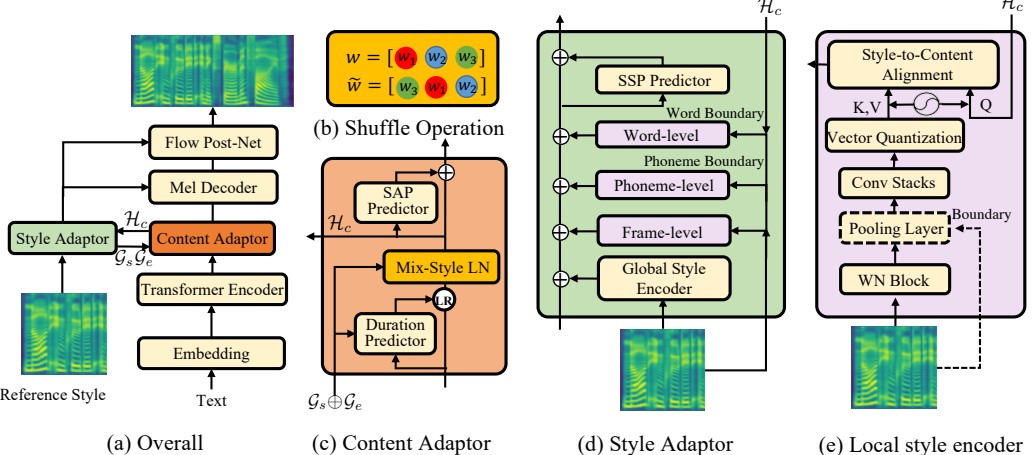

Figure 1: Architecture of GenerSpeech. In subfigure (b), we use $w$ and $\tilde{w}$ to denote the input and output of the shuffle operation. In subfigure (c) and (d), we use the sinusoidal-like symbol to denote the positional encoding. LR: length regulator, LN: layer normalization, SAP: style-agnostic pitch, SSP: style-specific pitch. In subfigure (e), the operations denoted with dotted lines are included except the frame-level style encoder.

## 3.3 Generalizable Content Adaptor

To prevent the degradation in the style transfer from utterances with out-of-domain (OOD) custom voice, we eliminate the style information in phonetic sequences with the proposed Mix-Style Layer Normalization and predict the style-agnostic prosodic variations.

### 3.3.1 Mix-Style Layer Normalization

Previous work [1] found that layer normalization could greatly influence the hidden activation and final prediction with a light-weight learnable scale vector $\gamma$ and bias vector $\beta$: $\text{LN}(x) = \gamma\frac{x-\mu}{\sigma} + \beta$, where $\mu$ and $\sigma$ are the mean and variance of hidden vector $x$. [15, 6] further proposed conditional layer normalization for speaker adaptation $\text{CLN}(x, w) = \gamma(w)\frac{x-\mu}{\sigma} + \beta(w)$, which can adaptively perform scaling and shifting of the normalized input features based on the style embedding. Here two simple linear layers $E^\gamma$ and $E^\delta$ take style embedding $w$ as input and output the scale and bias vector respectively:

$$\gamma(w) = E^\gamma * w, \quad \beta(w) = E^\delta * w \tag{1}$$

The custom discrepancy between the source and target domain generally hinders the generalization capability of learned text-to-speech models. For disentangling style information and learning style-agnostic representation, a straightforward solution is to refine the sequence conditioned on the mismatched style information, which could be regarded as injecting noise to confuse the model and prevent it from generating style-consistent representation. Leveraging recent progress on domain generalization [9, 19, 53], in this work, we design the Mix-Style Layer Normalization for regularizing TTS model training by perturbing the style information in training samples:

$$\gamma_{\text{mix}}(w) = \lambda\gamma(w) + (1-\lambda)\gamma(\tilde{w}) \qquad \beta_{\text{mix}}(w) = \lambda\beta(w) + (1-\lambda)\beta(\tilde{w}), \tag{2}$$

where $w$ denotes the style vector, and $\tilde{w}$ is simply obtained by $\tilde{w} = \text{Shuffle}(w)$ (See Figure 6(b) for an illustration). where $\lambda \in \mathbb{R}^B$ are sampled from the Beta distribution, and $B$ represents the batch size. $\lambda \sim \text{Beta}(\alpha, \alpha)$ with $\alpha \in (0, \infty)$ being a trade-off between the original style and shuffle style, and we set $\alpha = 0.2$ throughout this paper. In the end, the generalizable style-agnostic hidden representations become:

$$\text{Mix-StyleLN}(x, w) = \gamma_{\text{mix}}(w)\frac{x - \mu}{\sigma} + \beta_{\text{mix}}(w) \tag{3}$$

Consequently, the model refines the input features regularized by perturbed style and learns generalizable style-invariant content representation. To further ensure diversity and avoid over-fitting, we perturb the style information by randomly mixing the shuffle vectors with a shuffle rate $\lambda$ sampled from the Beta distribution.

See Algorithm 1 in Appendix B for the PyTorch-like pseudo-code. In the final part, we adopt a pitch predictor to generate style-agnostic prosodic variations. By utilizing the Mix-Style Layer Normalization in the generalizable content adaptor, the linguistic content-related variation could be disentangled from the global style attributes (i.e., speaker and emotion), which promotes the generalization of TTS model towards out-of-domain custom style.

### 3.4 Multi-level Style adaptor

The out-of-domain custom voice generally contains high dynamic style attributes (e.g., speaker identities, prosodies, and emotions), making the TTS model difficult to model and transfer. As shown in Figure 6(d), we propose a multi-level style adaptor for both global and local stylization.

#### 3.4.1 Global Representation

We use a generalizable wav2vec 2.0 model [2] to capture the global style characteristics, including the speaker and emotion acoustic conditions. Wav2vec 2.0 is a recently proposed self-supervised framework for speech representation learning, which follows a two-stage training process of pre-training and finetuning, and has been demonstrated for its efficiency in learning discriminative embedding. In practice, we add an average pooling layer and fully-connected layers on the top of the wav2vec 2.0 encoder, which allows for fine-tuning the model on speaker and emotion classification tasks. The AM-softmax [47] criteria is employed as the loss function for downstream classification. Due to a limited amount of multi-speaker and multi-emotion speech data, we observe that fine-tuning the global encoders separately using different corpus could be better choice.

To sum up, the fine-tuned wav2vec 2.0 model generates discriminative global representations $\mathcal{G}_s$, and $\mathcal{G}_e$ to model the speaker and emotion characteristics, respectively. We put more detailed information about the wav2vec 2.0 model in Appendix A.2.1.

#### 3.4.2 Local Representation

To catch the fine-grained prosodic details, we consider the frame, phoneme, and word-level three differential acoustic conditions. These multi-level style encoders share a common architecture: First, the input sequences pass through several convolutional layers to get refined. Optionally, we conduct pooling operation on the refined series for different level stylization. In practice, the pooling operation averages the hidden states inside each representation according to the input boundary. Later, the refined sequences are fed into the vector quantization later as a bottleneck [45] to eliminate the non-prosodic information effectively. We describe these three-level prosodic conditions as follows:

1) **Frame level**. To catch the frame-level latent representation $\mathcal{S}_p$, we remove the optional pooling layer in the local style encoder. 2) **Phoneme level**. Considering the rises and falls of the pitch and stress, style patterns between phonemes could be highly dynamic. To catch the phoneme-level style latent representation $S_p$ from speech, we take the phoneme boundary as an extra input and apply pooling on the refined sequences before feeding into the vector quantization layer. 3) **Word level**. Similar to the phoneme-level stylization, the acoustic conditions (e.g., pitch and stress) on each word are highly variable. To catch the word-level style latent representation $S_w$ from speech, we take the word boundary as an extra input and apply pooling to refine the sequences.

In the **Vector Quantization** block, the refined sequences are fed into the vector quantization later as a bottleneck [45] to eliminate the non-style information effectively. The vector quantization block enjoys a carefully-crafted information bottleneck design. we define a latent embedding space $e \in R^{K \times D}$ where $K$ is the size of the discrete latent space (i.e., a $K$-way categorical), and $D$ is the dimensionality of each latent embedding vector $e_i$. Note that there are $K$ embedding vectors

$e_i \in \mathbb{R}^D, i \in 1, 2, \ldots, K$. To make sure the representation sequence commits to an embedding and its output does not grow, we add a commitment loss following previous work [45]:

$$\mathcal{L}_c = \|z_e(x) - \text{sg}[e]\|_2^2, \tag{4}$$

where $z_e(x)$ is the output of the vector quantization block, and sg stands for the stop gradient operator.

### 3.4.3 Style-To-Content Alignment Layer

To align the variable-length local style representations (i.e, $\mathcal{S}_u$, $\mathcal{S}_p$, and $\mathcal{S}_w$) with the phonetic representations $\mathcal{H}_c$, we introduce the Style-To-Content Alignment Layer for learning the alignment between the two modalities of style and content, which is illustrated in Figure 6(e).

In practice, we adopt the popular Scaled Dot-Product Attention [46] as the attention module. Taking the module in frame-level style encoder as an example, where $\mathcal{H}_c$ is used as the query, and $\mathcal{S}_u$ is used as both the key and the value:

$$\text{Attention}(Q, K, V) = \text{Attention}(\mathcal{H}_c, \mathcal{S}_u, \mathcal{S}_u) = \text{Softmax}(\frac{\mathcal{H}_c \mathcal{S}_u^T}{\sqrt{d}})\mathcal{S}_u \tag{5}$$

We add a positional encoding embedding to the style representations before they are fed into the attention module. For efficient training, we use a residual connection [10] to add the $\mathcal{H}_c$. A large dropout rate is adopted in the dropout layer to prevent the aligned representation from being directly copied from the linguistic content representations.

For better performance, we stack the style-to-content alignment layer multiple times and gradually stylize the query value (i.e., linguistic content representations). In the end, we utilize the pitch predictor to generate the style-specific prosodic variations.

## 3.5 Flow-based Post-Net

Expressive custom voices usually contain rich and high dynamic variation, while it's difficult for the widely-applied transformer decoder to generate such detailed mel-spectrogram samples. To further improve the quality and similarity of synthesized mel-spectrograms, we introduce a flow-based post-net to refine the coarse-grained outputs of the mel-spectrogram decoder.

The architecture of the post-net follows the Glow [21] family, which is conditioned on a coarse-grained spectrogram and the mel decoder input. During training, the flow post-net converts the synthesized mel-spectrogram into the gaussian prior distribution and calculates the exact log-likelihood of the data. During inference, we sample the latent variables from the prior distribution and pass them into the post-net reversely to generate the expressive mel-spectrogram.

## 3.6 Pre-training, Training and Inference Procedures

### 3.6.1 Pre-training and Training

In the pre-training stage of GenerSpeech, we finetune the global style encoder wav2vec 2.0 model to downstream tasks using the AM soft-max loss objective. All parameters are adjustable during this stage, after which we transfer the knowledge of the discriminatively-trained wav2vec 2.0 model to generate global style features.

In training GenerSpeech, the reference and target speech remain the same. The final loss terms consist of the following parts: 1) duration prediction loss $\mathcal{L}_{dur}$: MSE between the predicted and the GT phoneme-level duration in log scale; 2) mel reconstruction loss $\mathcal{L}_{mel}$: MAE between the GT mel-spectrogram and that generated by the transformer decoder; 3) pitch reconstruction loss $\mathcal{L}_p$: MSE between the GT and the joint pitch spectrogram predicted by the style-agnostic and style-specific pitch predictor. More details on pitch prediction have been included in Appendix A.2.2. 4) the negative log-likelihood of the post-net $\mathcal{L}_{pn}$; 5) commit loss $\mathcal{L}_c$: the objective to constrain vector quantization layer according to Equation 4. Also, we put details on training stability in Appendix D.

### 3.6.2 Inference

GenerSpeech conducts style transfer of custom voice for out-of-domain text-to-speech synthesis in the following pipeline: 1) The text encoder encodes the phoneme sequence, and the expanded representations $\mathcal{H}_c$ could be obtained according to the inference duration, following which the style-agnostic pitch (SAP) predictor generate the linguistic content speech variation invariant to custom style. 2) Given reference speech samples, we can obtain the word and phoneme boundaries by forced alignment, which are fed into the multi-level style adaptor to model the style latent representations: The wav2vec 2.0 model generates speaker $\mathcal{G}_s$ and emotion $\mathcal{G}_e$ representations to control the global style, and the local style encoder catches the frame, phoneme and word-level fine-grained style representations $\mathcal{S}_u$, $\mathcal{S}_p$, and $\mathcal{S}_w$, respectively. And then the style-specific pitch (SSP) predictor generates the style-sensitive variation. 3) The mel decoder generates the coarse-grained mel-spectrograms $\tilde{M}$, following which the flow-based post-net converts randomly sampled latent variables into the fine-grained mel-spectrograms $M$ conditioned on $\tilde{M}$ and the mel decoder input.

## 4 Experiments

### 4.1 Experimental Setup

#### 4.1.1 Dataset

In the pre-training stage, we adopt the multi-emotion dataset IEMOCAP [35] which contains about 12 hours of emotional speech, and the multi-speaker dataset VoxCeleb1 [32] which contains over 100,000 utterances from 1,251 celebrities. In the training stages, we utilize the LibriTTS [33] dataset, which is a multi-speaker corpus (2456 speakers) derived from LibriSpeech [49] and contains 586 hours of speech data. Additionally, we use part of the ESD database [54] which consists of 13 hours of speech from 10 speakers with 3 emotions (angry, happy, and neutral) to include more acoustic variation. To evaluate GenerSpeech in the out-of-domain scenario, we generalize the source model to other datasets with different acoustic conditions from training data, including the VCTK dataset (a multi-speaker dataset with 108 unseen speakers) and the leaving part of the ESD database with 2 unseen emotions (surprise and sad). We randomly choose 20 sentences with speaker and emotion unseen during training to construct the out-of-domain (OOD) testing set.

Following the common practice [6, 31], we conduct preprocessing on the speech and text data: 1) convert the sampling rate of all speech data to 16kHz; 2) extract the spectrogram with the FFT size of 1024, hop size of 256, and window size of 1024 samples; 3) convert it to a mel-spectrogram with 80 frequency bins.

#### 4.1.2 Model Configurations

GenerSpeech consists of 4 feed-forward Transformer blocks for the phoneme encoder and mel-spectrogram decoder. We add a linear layer in a style adaptor to transform the 768-dimension global embedding from wav2vec 2.0 to 256 dimensions. The default size of the codebook in the vector quantization layer is set to 128. We stack multiple WaveNet layers in the style-to-content alignment attention block. We have attached more detailed information on the model configuration in Appendix A.

#### 4.1.3 Training and Evaluation

After the 100,000 pre-training steps, we train GenerSpeech for 200,000 steps using 1 NVIDIA 2080Ti GPU with a batch size of 64 sentences. Adam optimizer is used with $\beta_1 = 0.9, \beta_2 = 0.98, \epsilon = 10^{-9}$. We utilize HiFi-GAN[22] (V1) as the vocoder to synthesize waveform from the generated mel-spectrogram in all our experiments.

We conduct crowd-sourced human evaluations with MOS (mean opinion score) for naturalness and SMOS (similarity mean opinion score) [31] for style similarity on the testing set. Both metrics are rated from 1 to 5 and reported with 95% confidence intervals (CI). An AXY test [40] of style similarity is conducted to assess the style transfer performance, where raters are asked to rate a 7-point score (from -3 to 3) and choose the speech samples which sound closer to the target style in terms of style expression. We further include objective evaluation metrics: Speaker Cosine Similarity

Table 1: Quality and style similarity of parallel customization samples when generalized to out-of-domain VCTK and ESD testsets. The evaluation is conducted on a server with 1 NVIDIA 2080Ti GPU and batch size 1. The mel-spectrograms are converted to waveforms using Hifi-GAN (V1).

| Method | VCTK | | | | ESD | | | |
|---|---|---|---|---|---|---|---|---|
| | MOS | SMOS | Cos | FFE | MOS | SMOS | Cos | FFE |
| Reference | $4.40 \pm 0.09$ | / | / | / | $4.47 \pm 0.08$ | / | / | / |
| Reference(voc.) | $4.37 \pm 0.09$ | $4.30 \pm 0.09$ | 0.96 | 0.05 | $4.40 \pm 0.09$ | $4.47 \pm 0.10$ | 0.99 | 0.07 |
| Mellotron | $3.91 \pm 0.08$ | $3.88 \pm 0.08$ | 0.74 | 0.32 | $3.92 \pm 0.07$ | $4.01 \pm 0.08$ | 0.80 | 0.27 |
| FG-TransformerTTS | $3.95 \pm 0.1$ | $3.90 \pm 0.09$ | 0.86 | **0.30** | $3.90 \pm 0.10$ | $3.94 \pm 0.08$ | 0.67 | 0.43 |
| Expressive FS2 | $3.85 \pm 0.08$ | $3.87 \pm 0.10$ | 0.85 | 0.41 | $4.04 \pm 0.08$ | $3.93 \pm 0.09$ | 0.93 | 0.41 |
| Meta-StyleSpeech | $3.90 \pm 0.07$ | $3.95 \pm 0.08$ | 0.83 | 0.38 | $4.02 \pm 0.10$ | $3.97 \pm 0.10$ | 0.86 | 0.41 |
| Styler | $3.89 \pm 0.09$ | $3.82 \pm 0.08$ | 0.76 | 0.38 | $3.76 \pm 0.08$ | $4.05 \pm 0.08$ | 0.68 | 0.39 |
| GenerSpeech | **$4.06 \pm 0.08$** | **$4.01 \pm 0.09$** | **0.88** | 0.35 | **$4.11 \pm 0.10$** | **$4.20 \pm 0.09$** | **0.97** | **0.26** |

Table 2: The AXY preference test results for parallel and non-parallel style transfer. We select 20 samples from VCTK and ESD testing sets for evaluation. For each reference (A), the listeners are asked to choose a preferred one among the samples synthesized by baseline models (X) and proposed GenerSpeech (Y), from which AXY preference rates are calculated. The scale ranges of 7-point are from "X is much closer" to "Both are about the same distance" to "Y is much closer", and can naturally be mapped on the integers from -3 to 3.

| Baseline | Parallel | | | | Non-Parallel | | | |
|---|---|---|---|---|---|---|---|---|
| | 7-point score | Perference (%) | | | 7-point score | Perference (%) | | |
| | | X | Neutral | Y | | X | Neutral | Y |
| Mellotron | $1.51 \pm 0.10$ | 26% | 14% | 40% | $1.62 \pm 0.09$ | 6% | 28% | 66% |
| FG-TransformerTTS | $1.07 \pm 0.14$ | 22% | 30% | 48% | $1.29 \pm 0.10$ | 34% | 20% | 46% |
| Expressive FS2 | $1.22 \pm 0.12$ | 30% | 20% | 50% | $1.42 \pm 0.11$ | 24% | 16% | 60% |
| Meta-StyleSpeech | $1.13 \pm 0.09$ | 26% | 26% | 48% | $1.18 \pm 0.12$ | 14% | 26% | 60% |
| Styler | $1.49 \pm 0.10$ | 18% | 24% | 58% | $1.27 \pm 0.09$ | 20% | 22% | 58% |

(Cos) and F0 Frame Error (FFE) measure the timbre and prosody similarity among the synthesized and reference audio, respectively. More information on evaluation has been attached in Appendix C.

#### 4.1.4 Baseline models

We compare the quality and similarity of generated audio samples of our GenerSpeech with other systems, including 1) Reference, the reference audio; 2) Reference (voc.), where we first convert the reference audio into mel-spectrograms and then convert them back to audio using HiFi-GAN; 3) Mellotron [44]: The auto-regressive multi-speaker TTS model based on the Tacotron using global style token (GST). 4) FG-TransformerTTS [5]: The fine-grained style control on auto-regressive model Transformer-TTS. 5) Expressive FS2 [37]: The combination of both multi-speaker [7] and muli-emotion [8] FastSpeech 2, which adds the speaker and emotion d-vectors extracted by the pre-trained discriminative models to the backbone. 6) Meta-StyleSpeech [31]: The finetuned multi-speaker text-to-speech model with meta-learning. 7) Styler [24]: The expressive text-to-speech model that model style factor via speech decomposition.

### 4.2 Performance

We randomly draw audio samples from the OOD testing sets as references to evaluate GenerSpeech and baseline models towards style transfer for out-of-domain text-to-speech synthesis. And then, we synthesize speech using the reference audio and given arbitrary text. Considering the text consistency between the reference and generated speech samples, we could cluster our experiments into two categories [40]: **Parallel** and **Non-Parallel** style transfer.

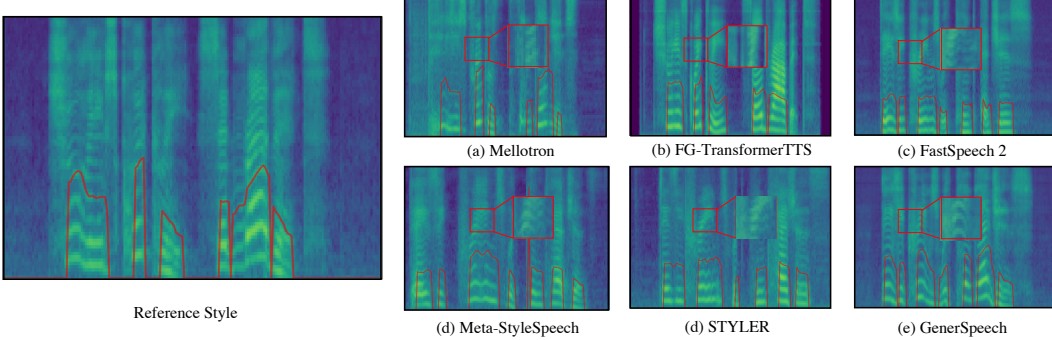

Figure 2: Visualizations of the reference and generated mel-spectrograms in Non-Parallel style transfer. The corresponding texts of reference and generated speech samples are "*Daisy creams with pink edges.*" and "*Chew leaves quickly, said rabbit.*", respectively.

### 4.2.1 Parallel Style Transfer

We first demonstrate that our model can generalize to OOD custom voices when the text is unchanged from the reference utterance. We calculate the objective matrixes (i.e., Cos, and FFE) between the generated and reference sample. For easy comparison, the results are compiled and presented in Table 1, and we have the following observations: 1) For audio quality, GenerSpeech has achieved the highest MOS with scores of $4.06$ (VCTK) and $4.11$ (ESD) compared with the baseline models, especially in ESD dataset. 2) In terms of style similarity, GenerSpeech score the highest overall SMOS of $4.01$ (VCTK) and $4.20$ (ESD). The objective results of both Cos and FFE further show that GenerSpeech surpasses the state-of-the-art models in transferring the style of custom voices. Informally, The proposed multi-level style adaptor allows GenerSpeech to generate speech samples that match the style of the out-of-domain reference substantially more accurately, clearly reflecting the correct gender, pitch, and formant ranges. We put more visualizations of mel-spectrograms towards parallel style transfer in Appendix F.

### 4.2.2 Non-Parallel Style Transfer

In this section, we explore the robustness of our proposed model in non-parallel style transfer, in which a TTS system synthesizes different text in the prosodic style of a reference signal. We randomly choose 20 reference signals from the testing set and test how TTS models replicate each style when synthesizing different target phrases.

We conduct a subjective evaluation to assess the style similarity of synthesized speech to reference one. As shown in Table 2, the side-by-side subjective test indicates that raters prefer GenerSpeech synthesis against baselines. The proposed multi-level style adaptor significantly improves GenerSpeech to inform the speech style, allowing an expressive reference sample to guide the robust stylistic synthesis of arbitrary text.

We further plot the mel-spectrograms and corresponding pitch tracks generated by the TTS systems in Figure 2, and have the following observations: 1) GenerSpeech can generate mel-spectrograms with rich details in frequency bins between two adjacent harmonics, unvoiced frames, and high-frequency parts, which results in more natural sounds. However, some baseline models (especially Mellotron) fail to generate high-fidelity mel-spectrograms in Non-Parallel style transfer; 2) GenerSpeech can resemble the prosodic style of the reference signal and demonstrates its precise style transfer, which is nearly time-aligned in pitch contours. However, most baseline models failed to match the prosodic style. They generated the "average" distribution over their input data, generating less expressive speech, especially for long-form phrases.

### 4.3 Ablation Study

As shown in Table 3, we conduct ablation studies to demonstrate the effectiveness of several designs in GenerSpeech, including the generalizable content adaptor, multi-level style adaptor, and the post-net. The global representations provide the basic timbre and emotion attributes so that we leave them unchanged. We conduct CMOS (comparative mean opinion score) and CSMOS (comparative similarity mean opinion score) evaluations and have the following findings: 1) Both the quality and similarity scores drop when removing the multi-level local style encoder, which demonstrates the efficiency of the proposed method in capturing style latent representations. 2) Replacing mix-style layer normalization in a generalizable content adaptor with the original one results in decreased qual-

Table 3: Audio quality and similarity comparisons for ablation study. LSE and MS-LN respectively represent the multi-level local style encoder and Mix-Style layer normalization.

| Setting | CMOS | CSMOS |
|---|---|---|
| GenerSpeech | 0.0 | 0.0 |
| w/o LSE | -0.02 | -0.15 |
| w/o MS-LN | -0.06 | -0.07 |
| w/o post-net | -0.10 | -0.02 |

ity and similarity, verifying the significance of mix-style calculation. 3) Removing the flow-based post-net has witnessed the degradation of audio quality, proving that the post-net could refine the coarse-grained output and generate spectrograms with increasing details. To demonstrate the effectiveness of fine-grained modeling, we attach the visualizations of mel-spectrograms in Appendix F.

## 4.4 Style Adaptation

Following previous work [6], we further study the performance of our model in style adaptation with different amounts of data. We randomly sample utterances (each around 5 sec) from the out-of-domain datasets and

Table 4: Adaptation performance with varying data.

| #Sample | 0 | 1 | 2 | 5 | 10 | 20 |
|---|---|---|---|---|---|---|
| CMOS | 0.00 | + 0.01 | + 0.01 | + 0.05 | + 0.07 | + 0.07 |
| CSMOS | 0.00 | + 0.03 | + 0.04 | + 0.10 | + 0.13 | + 0.14 |

finetune the parameters in the multi-level style adaptor for additional 2000 steps. The CMOS and CSMOS evaluation results have been illustrated in Table 4: GenerSpeech performs better with the increasing amount of adaptation data, demonstrating the ability of proposed GenerSpeech towards adaptive style transfer in the few-shot data setting. The detailed fine-tuning setting has been included in Appendix E.

## 5 Conclusion

In this work, we proposed GenerSpeech, a text-to-speech model towards high-fidelity zero-shot style transfer of out-of-domain custom voices. To achieve better model generalization, we design several techniques to learn the style-agnostic and style-specific variations in speech separately: 1) GenerSpeech utilized a multi-level style adaptor to model and transfer various style attributes, including the speaker and emotion global characteristics, and the fine-grained frame, phoneme, and word-level prosodic representations; 2) The Mix-Style layer normalization was further adopted to eliminate the style information in linguistic representations for improving the generalization of GenerSpeech. Experimental results demonstrated that GenerSpeech achieved new state-of-the-art zero-shot style transfer results for OOD text-to-speech synthesis. Our further extension study to adaptation further showed that GenerSpeech performed robustly in the few-shot data setting. For future work, we will further verify the effectiveness of GenerSpeech on more general scenarios such as multilingual generalization. We envisage that our work could serve as a basis for future text-to-speech synthesis studies.

## Acknowledgements

This work was supported in part by the National Natural Science Foundation of China (Grant No.62072397 and No.61836002), Zhejiang Natural Science Foundation (LR19F020006), Yiwise, and National Key R&D Program of China (Grant No.2020YFC0832505).

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
