# Appendices

## GenerSpeech: Towards Style Transfer for Generalizable Out-Of-Domain Text-to-Speech

## A Details of Models

In this section, we describe details in the phoneme encoder, generalizable content adaptor, multi-level style adaptor, flow-based post-net and the models.

### A.1 Model Configurations

We list the model hyper-parameters of GenerSpeech in Table 5.

| Hyperparameter | | GenerSpeech |
|---|---|---|
| Text Encoder | Phoneme Embedding | 192 |
| | Encoder Layers | 4 |
| | Encoder Hidden | 256 |
| | Encoder Conv1D Kernel | 9 |
| | Encoder Conv1D Filter Size | 1024 |
| | Encoder Attention Heads | 2 |
| | Encoder Dropout | 0.1 |
| Generalizable Content Adaptor | Style-Agnostic Pitch Predictor Conv1D Kernel | 3 |
| | Style-Agnostic Pitch Predictor Conv1D Filter Size | 256 |
| | Style-Agnostic Pitch Predictor Dropout | 0.5 |
| | Probability of using MixStyle | 0.2 |
| | Beta distribution parameter $\alpha$ | 0.1 |
| Multi-level Style Adaptor | Style-Specific Pitch Predictor Conv1D Kernel | 3 |
| | Style-Specific Pitch Predictor Conv1D Filter Size | 256 |
| | Style-Specific Pitch Predictor Dropout | 0.5 |
| | Multi-level Style Adaptor Hidden | 256 |
| | Local Style Encoder WN Layers | 4 |
| | Local Style Encoder VQ Codebook Size | 128 |
| | Local Style Encoder Conv Stack Layers | 5 |
| | Style-to-Content Alignment Layers | 2 |
| Mel-Spectrogram Decoder | Decoder Layers | 4 |
| | Decoder Hidden | 256 |
| | Decoder Conv1D Kernel | 9 |
| | Decoder Conv1D Filter Size | 1024 |
| | Decoder Attention Headers | 2 |
| | Decoder Dropout | 0.1 |
| Post-Net | WaveNet Layers | 3 |
| | WaveNet Kernel | 3 |
| | WaveNet Channel Size | 192 |
| | Flow Steps | 12 |
| | Shared Groups | 3 |
| Total Number of Parameters | | 51M |

Table 5: Hyperparameters of GenerSpeech models.

### A.2 Content and Style Adaptor

#### A.2.1 Global Style Encoder

As illustrated in Figure 3, the main body of the model consists of a CNN-based feature encoder, a Transformer-based context network and a quantization module. The wav2vec 2.0 model builds context

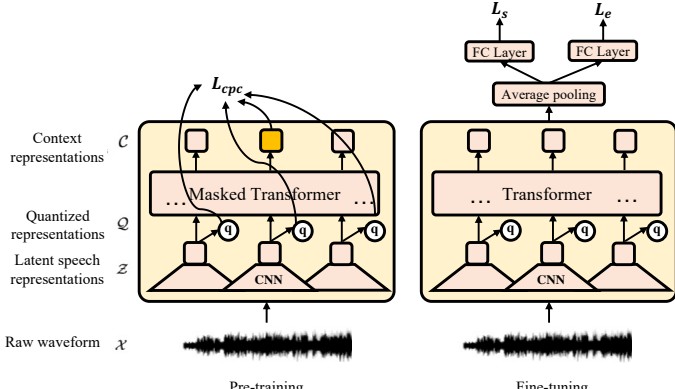

Figure 3: Illustration of the downstream global style encoder. The model architecture used in the pre-training stage and finetuning stage are identical, except for the quantization modules and extra output layers.

representations over continuous speech representations and self-attention captures dependencies over the entire sequence of latent representations end-to-end. We finetune the parameters of the w2v-encoder (around 94M). We add two fully connected layers on the top of the w2v-encoder to predict the speaker and emotion global latent representations in parallel.

### A.2.2 Pitch Prediction

The overall pitch prediction pipeline mainly follows previous non-autoregressive TTS models [37], except that GenerSpeech adopts two pitch predictors to generate style-specific and style-agnostic pitch spectrograms, respectively. As shown in Figure 3(b), the Style-Specific Pitch (SSP) predictor and Style-Agnostic Pitch (SAP) predictor enjoy the same architecture. During training, we add the output of the SSP predictor and SIP predictor to obtain the joint pitch spectrogram as illustrated in Figure 4. We train them with ground-truth pitch spectrogram and the mean/variance of pitch contour and optimize it with mean square error. We infer the style-specific and style-agnostic pitch spectrogram separately and inverse the joint pitch spectrogram to pitch contour with inverse continuous wavelet transform (iCWT).

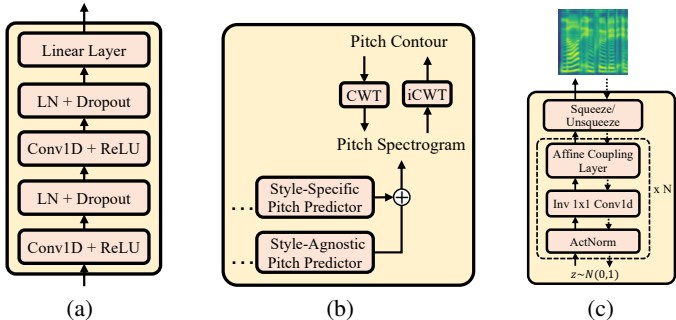

(a)                          (b)                          (c)

Figure 4: (a) The common architecture of pitch predictor. (b) Details in pitch prediction. CWT and iCWT denote continuous wavelet transform and inverse continuous wavelet transform respectively. (c)The flow-based post-net gets a mel-spectrogram and squeezes it, following which it gets processed through a number of flow blocks. Each flow block contains activation normalization layer, affine coupling layer, and invertible 1x1 convolution layer.

### A.3 Flow-based post-net

The flow-based models can overcome the over-smoothing problem and generate more realistic outputs. To model rich details in ground-truth mel-spectrograms, we introduce a flow-based post-net with

strong condition inputs to refine the coarse-grained mel-spectrogram. As shown in Figure 4(c), the flow-based post-net is composed of a family of flows that can perform forward and inverse transformation in parallel. During training, the post-net efficiently transforms a mel-spectrogram into the latent representation for maximum likelihood estimation. During inference, it transforms the prior distribution into the mel-spectrogram distribution efficiently to parallelly generate the high-quality sample.

# B    Pseudo-Code of Mix-Style Layer Normalization

Algorithm 1 provides a PyTorch-like pseudo-code.

---

**Algorithm 1** PyTorch-like pseudo-code for Mix-Style Layer Normalization.

```
# x: input features of shape (B, T, C)
# global_embed: tha element-wise addition of the speaker and emotion embedding (B, 1, C)
# p: probabillity to apply MixStyle
# alpha: hyper-parameter for the Beta distribution
# eps: a small value added before square root for numerical stability

if not in training mode:
    return x

if random probability > p:
    return x

B = x.size(0) # batch size

mu, sig = torch.mean(x, dim=-1, keepdim=True), torch.std(x, dim=-1, keepdim=True)
x_normed = (x - mu) / (sig + eps) # normalize input

lmda = Beta(alpha, alpha).sample((B, 1, 1)) # sample instance-wise convex weights
lmda = lmda.to(x.device)

# Get Bias and Gain
mu1, sig1 = torch.split(self.affine_layer(global_embed), self.hidden_size, dim=-1)

# MixStyle
perm = torch.randperm(B) # generate shuffling indices
mu2, sig2 = mu1[perm], sig1[perm] # shuffling

mu_mix = mu1 * lmda + mu2 * (1 - lmda) # generate mixed mean
sig_mix = sig1 * lmda + sig2 * (1 - lmda) # generate mixed standard deviation

# Perform Scailing and Shifting
return x_normed * sig_mix + mu_mix # denormalize input using the mixed statistics
```

---

# C    Evaluation

## C.1    Subjective Evaluation

For audio quality evaluation, we conduct the MOS (mean opinion score) tests and explicitly instruct the raters to "*(focus on examining the audio quality and naturalness, and ignore the differences of style (timbre, emotion and prosody).)*". The testers present and rate the samples, and each tester is asked to evaluate the subjective naturalness on a 1-5 Likert scale.

For style similarity evaluation, we explicitly instruct the raters to "*(focus on the similarity of the style (timbre, emotion and prosody) to the reference, and ignore the differences of content, grammar, or audio quality.)*". In the SMOS (similarity mean opinion score) tests, we paired each synthesized utterance with a ground truth utterance to evaluate how well the synthesized speech matches that from the target speaker. Each pair is rated by one rater. In the AXY discrimination test, a human rater is presented with three stimuli: a reference speech sample (A), and two competing samples (X and Y) to evaluate. The rater is asked to rate whether the prosody of X or Y is closer to that of the reference on a 7-point scale. The scale ranges from "X is much closer" to "Both are about the same distance" to "Y is much closer", and can naturally be mapped on the integers from -3 to 3.

In the ablation study, we further conduct CMOS (comparative mean opinion score) and CSMOS (comparative similarity mean opinion score) evaluations. Listeners are asked to compare pairs of audio generated by systems A and B, indicate which of the two audio they prefer, and choose one of

the following scores: 0 indicating no difference, 1 indicating small difference, 2 indicating a large difference, and 3 indicating a very large difference.

Our subjective evaluation tests are crowd-sourced and conducted by 25 native speakers via Amazon Mechanical Turk. The screenshots of instructions for testers have been shown in Figure 5. We paid $8 to participants hourly and totally spent about $750 on participant compensation. A small subset of speech samples used in the test is available at `https://GenerSpeech.github.io/`.

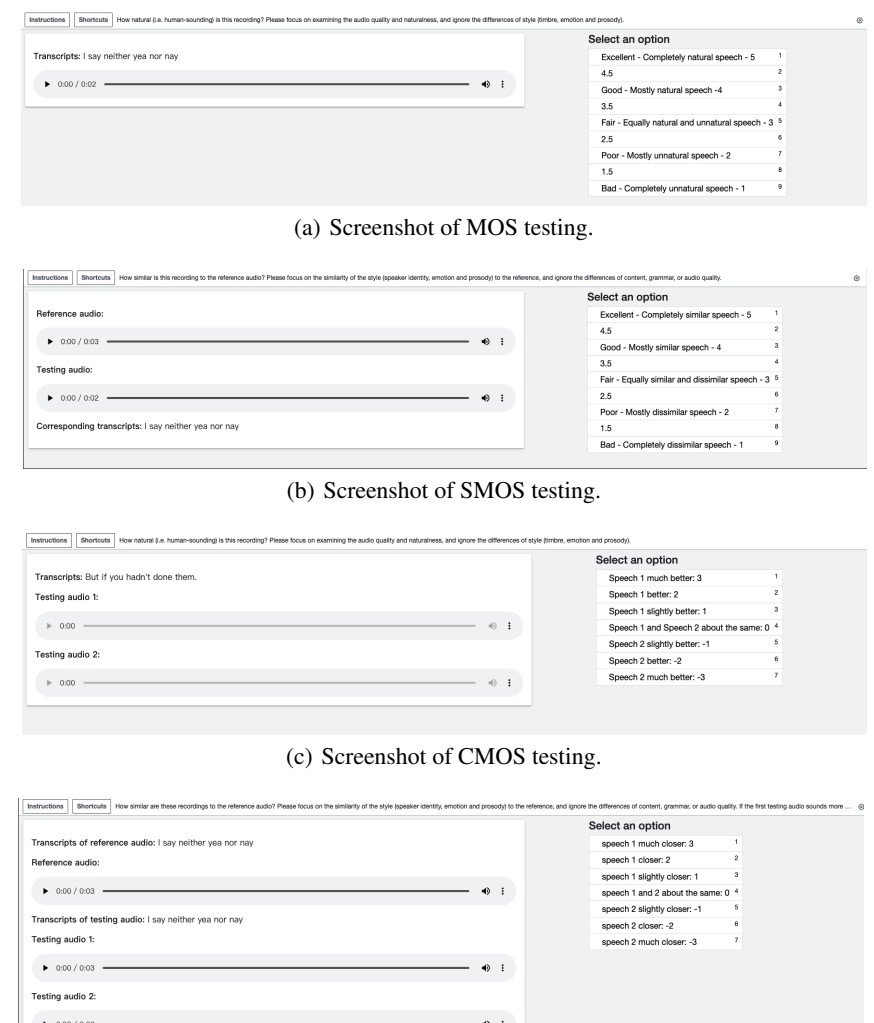

(a) Screenshot of MOS testing.

(b) Screenshot of SMOS testing.

(c) Screenshot of CMOS testing.

(d) Screenshot of AXY discrimination and CSMOS testing.

Figure 5: Screenshots of subjective evaluations.

## C.2 Objective Evaluation

Cosine similarity is an objective metric that measures speaker similarity among multi-speaker audio. We compute the average cosine similarity between embeddings extracted from the synthesized and ground truth embeddings to measure the speaker similarity performance objectively.

F0 Frame Error (FFE) combines voicing decision error and F0 error metrics to capture F0 information.

## D Training Stability

It is well known that vector quantization tends to suffer from index collapse [36], which limits the expression ability of the proposed multi-level style encoder and hurts learning the style-to-content

alignment. To improve training stability, we empirically find that a warm-up strategy is efficient: 1) we remove the vector quantization layer and use hard force alignment in place of the learned style-to-content attention alignment in the first 20k steps. 2) after the first 20k steps, we add the vector quantization layer as the prosody bottleneck and the soft style-to-content alignment layer for later training.

## E  Fine-tuning

We fine-tune GenerSpeech using 1 NVIDIA 2080Ti GPU with the batch size of 64 sentences for 2000 steps, where the parameters of the whole model are optimized. The optimizer configuration and loss functions stay consistent with those in the experimental setup.

## F  More Visualization of Mel-Spectrograms

We put more visualizations of mel-spectrograms towards parallel style transfer for out-of-domain text-to-speech synthesis.

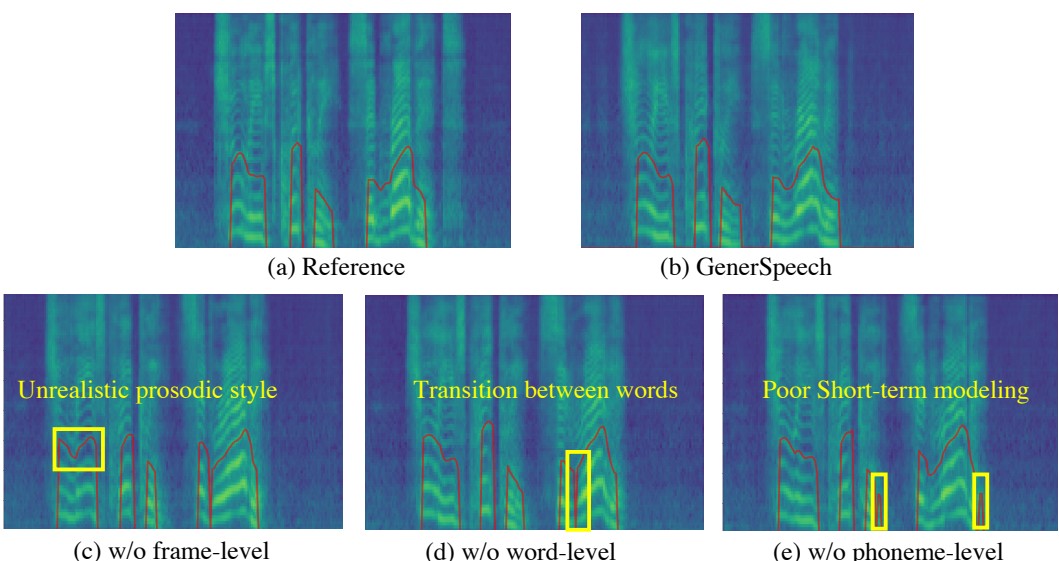

Figure 6: Visualizations of the ground-truth and generated mel-spectrograms in Parallel Style Transfer. The corresponding text is "*Chew leaves quickly, said rabbit.*".

## G  Visualization of Attention Weights

We put some attention visualizations in Figure 8. We can see that GenerSpeech can create reasonable alignments which are close to the diagonal in differential local-level style encoders, which helps the high-fidelity stylization.

## H  Potential Negative Societal Impacts

GenerSpeech lowers the requirements for high-quality and expressive text-to-speech synthesis, which may cause unemployment for people with related occupations such as broadcaster and radio host. In addition, there is the potential for harm from non-consensual voice cloning or the generation of fake media and the voices of the speakers in the recordings might be over-used than they expect.

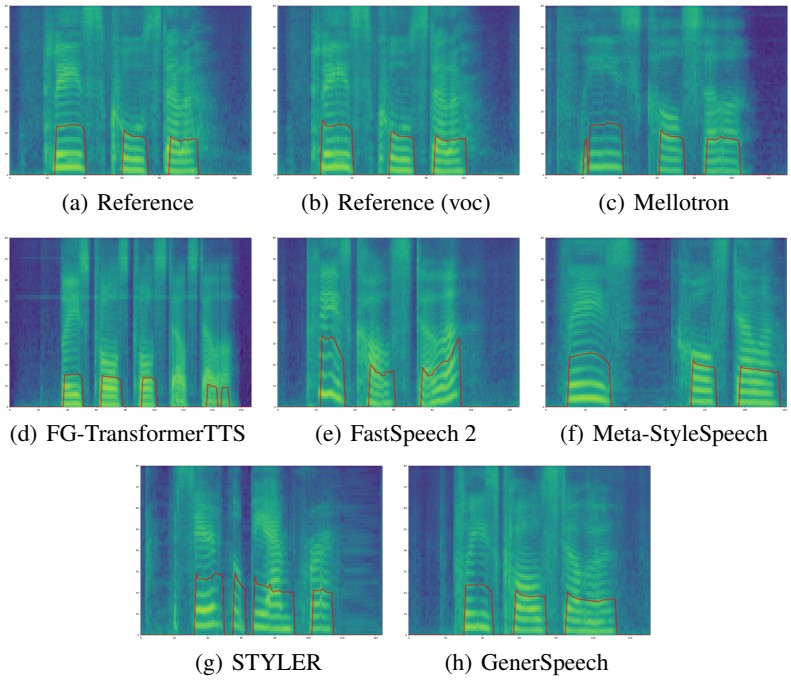

(a) Reference      (b) Reference (voc)      (c) Mellotron

(d) FG-TransformerTTS      (e) FastSpeech 2      (f) Meta-StyleSpeech

(g) STYLER      (h) GenerSpeech

Figure 7: Visualizations of the ground-truth and generated mel-spectrograms in Parallel Style Transfer. The corresponding text is "*Please call Stella.*".

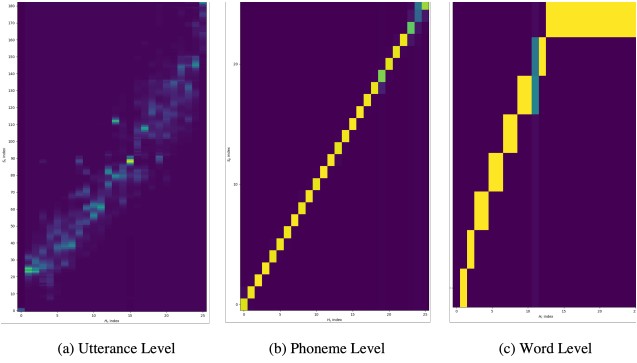

(a) Utterance Level      (b) Phoneme Level      (c) Word Level

Figure 8: Visualizations of the attention weights in parallel style transfer. The corresponding texts is "*Chew leaves quickly, said rabbit.*".

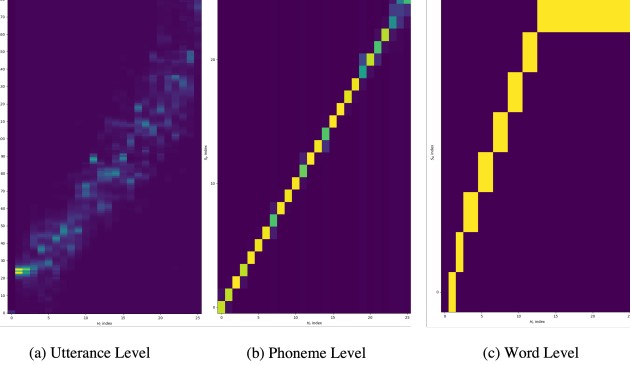

(a) Utterance Level      (b) Phoneme Level      (c) Word Level

Figure 9: Visualizations of the attention weights in non-parallel style transfer. The corresponding texts of reference and generated speech samples are "*Daisy creams with pink edges.*" and "*Chew leaves quickly, said rabbit.*", respectively.