# OpenReview forum: "GenerSpeech: Towards Style Transfer for Generalizable Out-Of-Domain Text-to-Speech"
_NeurIPS.cc/2022/Conference — NeurIPS 2022 Accept_

### Official Review · Reviewer_Mo8N · 2022-07-05

**Rating:** 6
**Confidence:** 5
**Soundness:** 3 good
**Presentation:** 3 good
**Contribution:** 3 good

**Summary:**

This work proposed a non-autoregressive TTS model with good style transfer under out of domain conditions. It includes 1) a multi-level style adaptor for global styles (speaker and emotion) and local styles (utterance, phoneme and word) 2) a generalizable content adaptor with mix-style layer-normalization 3) a flow based post-net. The experimental results show the proposed method could outperform baselines compared with and demonstrate the efficacy of the proposed method.

**Questions:**


 After listening the sample audio clips, one impression I have the style transfer from the expressive FS2 is as good as the proposed GenerSpeech. One difference I noticed is the synthesized audio quality. Could we add another experiment to compare the expressive FS2 with a flow-based post-net?

What is the shape of $Su$ after VQ? Is it a vector or a sequence of embedding vectors? If it is a sequence of embedding vectors, is there any constraint applied to make them identical or close?

The local multi-layer style encoders share the common architecture, are they all use VQ code size=128? How do we choose this number?

What's the purpose to include positional encoding embedding to the style representation before they are fed into the attention module?

**Ethics Review Area:**

["I don’t know"]

**Limitations:**

The authors have addressed that in section 5.

**Strengths And Weaknesses:**

Strengths:
1. Proposing a layer-normalized mix-style for better generalization
2. A style-specific module to integrate different style information.
3. Including a flow-based post-net on the top of the fast speech 2 model
4. Detailed comparison with different baselines.

Weaknesses:
1. Despite the authors' lengthy appendix, which provides more information about the experiments,  some critical details of the proposed method are still missing. For example, what is the shape of $Su$ after VQ? Is it a vector or a sequence of embedding vectors? If it is a sequence of embedding vectors, is there any constraint applied to make them identical or close?
2. Some abbreviations are used without definition, such as USE, PSE and WSE in Table 3.

---

> ### Author Response · Authors · 2022-08-02
> **Response to Reviewer Mo8N**
>
> We are grateful for your positive review and valuable feedback, and we hope our response fully resolves your concern.
>
> **[About the style representation $\mathcal{S_u}$.]**
>
> The style representation $\mathcal{S_u}$ is a sequence of embedding vectors. In style encoders, the vector quantization block serves as a bottleneck to eliminate the style-unrelated information effectively, which is regularized by the gradient of pitch reconstruction loss $\mathcal{L_p}$ in SSP Predictor. To further ensure the representation sequence does not explode, we include a commitment loss $\mathcal{L_c}$ as described in Section 3.6.1.
>
> **[About the abbreviations in Table 3.]**
>
> We use USE, PSE, and WSE to denote the utterance, phoneme, and word-level style encoder, respectively, which have been presented in the caption of Table 3.
>
> **[About the comparison with expressive FS2.]**
>
> Thanks for the reviewer's suggestion. We further include the expressive FS2 with post-net for comparison in the ESD dataset. The evaluation procedure stays consistent with the manuscript, and we present the results of parallel style transfer in the following tables:
>
> Method | MOS | SMOS | Cos | FFE
> - | -  | -| - | -
> Reference | 4.47 $\pm$ 0.08 | / | / | /
> Reference(voc.)  | 4.40 $\pm$ 0.09 | 4.47 $\pm$ 0.10 | 0.99 | 0.07
> Expressive FS2 | 4.04 $\pm$ 0.08  | 3.93 $\pm$ 0.09  | 0.93 |  0.41
> Expressive FS2 + Post-Net  | 4.09 $\pm$ 0.08   | 3.95 $\pm$ 0.08  | 0.94  | 0.39
> GenerSpeech   | **4.11 $\pm$ 0.10**  | **4.20 $\pm$ 0.09**  |  **0.97**  | **0.26**
>
> The flow-based post-net is designed to refine the coarse-grained outputs of the mel-spectrogram decoder, and thus an improvement in audio quality and naturalness could be observed. Regarding style similarity, the expressive FS2 with post-net shares a commonly limited capability with the original FS2 in modeling the highly dynamic style variation, showing an apparent gap from GenerSpeech in SMOS and FFE evaluation.
>
> To conclude, the performance gap of samples generated between the expressive FS2 and GenerSpeech is mainly attributed to the different capacities in modeling fine-grained style patterns (e.g., local rises and falls of the pitch and stress). We illustrate the pitch tracks of generated mel-spectrograms in Fig. 2 and find that GenerSpeech precisely resembles and transfers the prosodic style of a reference signal, which is nearly time-aligned in pitch contours. In contrast, expressive FS2 tends to model the "average" prosodic distribution over their input data, generating less expressive speech especially for long-form phrases.
>
> **[About the size of VQ code-book.]**
>
>
> In response to the reviewer's question, the vector quantization block enjoys a carefully-crafted information bottleneck design. We conducted ablation studies before deciding the optimal choice of the VQ size, and the results are presented in the following tables:
>
> Method | SMOS
> - | :-:
> Reference        | 4.47 $\pm$ 0.08
> Reference(voc.)   | 4.40 $\pm$ 0.09
> GenerSpeech(VQ Size=64)     | 4.02 $\pm$ 0.08
> GenerSpeech(VQ Size=96)     | 4.06 $\pm$ 0.07
> GenerSpeech(VQ Size=128)     | **4.11 $\pm$ 0.10**
> GenerSpeech(VQ Size=160)     | 4.05 $\pm$ 0.09
>
>
> GenerSpeech with a 64-category code-book has witnessed a decreased sample similarity, demonstrating that the tighter latent space fails to represent the diverse style patterns. In contrast, an expanded code-book (e.g., 160) produces "information leakage" where the content information of reference audio is unexpectedly modeled in the style encoder, and thus the entangled representation leads to distinct quality degradation. As a result, we set VQ size as 128, which is robust across style encoders in multiple levels.
>
> **[About the positional encoding embedding.]**
>
> To make use of the order of the style representation sequence, we introduce the positional encoding embedding to include information about the position. Consequently, it creates reasonable alignments close to the diagonal in differential local-level style encoders, properly controlling and transferring the prosodic variations in different places.
>
>
> Again, we appreciate your positive reviews and hope our response can fully resolve your concern.

---

> ### Author Response · Authors · 2022-08-09
> **Looking forward to further feedback**
>
> Dear Reviewer Mo8N,
>
> Thanks again for your constructive comments. We would like to kindly remind you that we tried our best to respond to your concerns with additional experiments, etc. As the end of the author-reviewer discussion period is approaching, we would be grateful if we could hear your feedback regarding our answers to the reviews. We would be happy to answer and discuss if you have further comments.
>
> Best regards, Authors

---

### Official Review · Reviewer_X6YQ · 2022-07-11

**Rating:** 5
**Confidence:** 4
**Soundness:** 2 fair
**Presentation:** 3 good
**Contribution:** 3 good

**Summary:**

This paper proposed GenerSpeech, which decomposed the speech variation into the style-agnostic and style-specific parts by introducing a multi-level style adaptor modeling both global speaker and emotion characteristics and the local fine-grained prosodic representations, and a generalizable content adaptor with Mix-Style Layer Normalization to eliminate style information in the linguistic content representation. The evaluations on style transfer demonstrated that GenerSpeech could synthesize high-quality speech in terms of audio quality and style similarity.

Update: the authors addressed my concerns well. I changed my ratings.

**Questions:**

In section 3.6.1, is the pitch reconstruction loss L_p only used to train the parameters in pitch predictors, or the gradient from pitch reconstruction loss is  also used to train the rest part of the model?

In section 4.4, when performing model adaptation using different amount of data, what's the detailed fine-tuning settings for the multi-level style adaptor? For example, what are the optimization configuration, and loss functions (if there are any changes).

**Limitations:**

The authored mentioned future research directions in section 5 and potential negative societal impacts in Appendix G.

**Strengths And Weaknesses:**

The introduced GenerSpeech model achieved better model generalization with several techniques to learn both the style-agnostic and style-specific variations in speech separately. The multi-level style adaptor could model and transfer various style attributes, including the speaker and emotion global characteristics, and the fine-grained utterance-level, phoneme-level, and word-level prosodic representations. The Mix-Style layer normalization was able to eliminate the style information in linguistic representations for improving the model generalization. GenerSpeech showed good performance in zero-shot style transfer results for OOD text-to-speech synthesis compared to other baseline methods. However, there are missing analyses about the effectiveness of the proposed style-agnostic and style-specific modules.

First, the author used shuffle operation in the Mix-Style Layer Normalization (MSLN) layer to eliminate the style information from linguistic content representation. But the study of the effectiveness of shuffle operation is missing, then how to guarantee the style information is fully disentangled and removed from the linguistic content representation by using shuffle in MSLN? Second, what's the latent representation extracted by the Style Adaptor from reference audio, such as different levels (utterance, word, phoneme) of latent representations? Do these local modules effectively capture the desired style information?

Minor: in Figure 1(e), it's helpful to mark where the Q(query) in "Style-to-Content Alignment" module comes from. Since 1(d) shows that the "Local style encoder" only takes mel-spectrogram as input.

---

> ### Author Response · Authors · 2022-08-02
> **Response to Reviewer X6YQ (1/2)**
>
> We thank the reviewer for the constructive feedback and for considering our work as “showed good performance” and “achieved better model generalization”. We understand that your concerns are mainly related to the paper's clarity and hope our response resolves your concerns fully.
>
> **[About the proposed mix-style layer normalization (MSLN).]**
>
> Thanks for the reviewer's feedback that requests more explanations about our proposed method. Conditional layer normalization has demonstrated its effectiveness in influencing the hidden activation and final prediction. AdaSpeech [1] and Meta-StyleSpeech [2] utilize the speaker embedding as the conditional information, which adaptively scales and shifts the normalized input features to synthesize speech of various voices.
>
>
> For disentangling style information and learning style-agnostic representation, a straightforward solution is to refine the sequence conditioned on the mismatched style information, which could be regarded as injecting noise to confuse the model and prevent it from generating style-consistent representation. Consequently, the model refines the input features regularized by perturbed style and learns generalizable style-invariant content representation. To further ensure diversity and avoid over-fitting, we perturb the style information by randomly mixing the shuffled vectors with a shuffle rate $\lambda$ sampled from the Beta distribution.
>
> From a high-level perspective described in Section 4.3, we observe that removing the mix-style layer normalization in a generalizable content adaptor results in decreased quality and similarity. Further, we conduct a toy experiment to verify the effectiveness of MSLN in disentangling the style information. Specifically, we fine-tune the learned phonetic representation $\mathcal{H}_c$ in the downstream speaker and emotion classification tasks using the LibriTTS (2456 speakers) and ESD (5 emotions) datasets, respectively. As the classifiers are converged, we compare the accuracy across the test set and present the results in the following tables:
>
> Method | Speaker Acc | Emotion Acc
> - | :-: | :-:
> $\mathcal{H}_c$ | 15.0\% |  34.5\%
> $\mathcal{H}_c$ with MSLN | 6.5\% | 25.0\%
>
>
> The phonetic representation $\mathcal{H}_c$ with MSLN has witnessed a distinct decrease in accuracy in the downstream classification tasks, getting close to random prediction. By introducing MSLN to the generalizable content adaptor, the global style attributes (i.e., speaker and emotion) could be disentangled from the linguistic content representation, which promotes the generalization of the TTS model towards out-of-domain custom voices.
>
>
> **[About the multi-level style adaptor.]**
>
> Thanks for the reviewer's feedback that requests more explanations about our proposed method. In the global style encoder, the speaker and emotion conditions have been constructed by a generalizable wav2vec 2.0 model, which mainly represents the overall style characteristics of a speech. However, considering the rises and falls of the local pitch and highly dynamic prosodic variations in custom voices, the fine-grained style representations should be adequately modeled.
>
> From a high-level point of view as described in Section 4.3, we observe a distinct drop when removing utterance, phoneme, or word-level style encoder, which demonstrates the effectiveness of capturing style latent representations in different receptive levels.
>
> Furthermore, we detail our analysis and investigation of different levels of style representation. Please kindly refer to Appendix F in the supplementary material, where we plot the mel-spectrograms and corresponding pitch tracks generated by different TTS systems: 1) The utterance-level latent representation mainly resorts to the long-term patterns with a large receptive field. Dropping the utterance-level style encoder results in an unrealistic prosodic style, making it more challenging to capture the long-term dependencies; 2) The phoneme level style encoder is supposed to model the short-term prosodic attributes between phonemes. Removing it has witnessed the incorrect fluctuation in local pitch, which indicates the effectiveness of the pooling operation in capturing short-term style variations. 3) A word may consist of several phonemes, and we find that removing the word-level style representation leads to an unnatural transition between words, resulting in a distinct drop in audio similarity.

---

> > ### Author Response · Authors · 2022-08-02
> > **Response to Reviewer X6YQ (2/2)**
> >
> >
> >
> > **[About the caption and mark in Fig. 1(e).]**
> >
> > Thanks for the reviewer's suggestion. We have detailed $Q$ in the style-to-content alignment module in the revised version of the paper.
> >
> > **[About the pitch reconstruction loss.]**
> >
> > The gradient of pitch reconstruction loss is utilized to optimize the whole model and prevents sub-optimal training.
> >
> > **[About the detailed fine-tuning setting.]**
> >
> > Following the common practice [1], we fine-tune GenerSpeech using 1 NVIDIA 2080Ti GPU with the batch size of 64 sentences for 2000 steps, and all parameters are optimized. The optimizer configuration and loss functions stay consistent with those in the experimental setup.
> >
> >
> > **[References]**
> >
> >
> > [1] Chen M, Tan X, Li B, et al. Adaspeech: Adaptive text to speech for custom voice[J]. ICLR, 2021.
> >
> > [2] Min D, Lee D B, Yang E, et al. Meta-stylespeech: Multi-speaker adaptive text-to-speech generation[C]//International Conference on Machine Learning. PMLR, 2021: 7748-7759.
> >
> > Again, we appreciate the reviewer's valuable reviews and believe some misunderstandings are due to our clarity. Hope our response can address your concerns.

---

> ### Author Response · Authors · 2022-08-09
> **Looking forward to further feedback**
>
> Dear Reviewer X6YQ,
>
> Thanks again for your constructive comments. We would like to kindly remind you that we tried our best to respond to your concerns with additional experiments, etc. As the end of the author-reviewer discussion period is approaching, we would be grateful if we could hear your feedback regarding our answers to the reviews. We would be happy to answer and discuss if you have further comments.
>
> Best regards, Authors

---

> ### Author Response · Authors · 2022-08-10
> **Thanks for raising your score!**
>
> We greatly appreciate that you have raised your score. We believe that your valuable comments have improved the paper, and feel free to ask more questions if you have any time. Thank you again for raising the score.

---

### Official Review · Reviewer_HPwf · 2022-07-12

**Rating:** 7
**Confidence:** 4
**Soundness:** 3 good
**Presentation:** 3 good
**Contribution:** 3 good

**Summary:**

The paper addresses the problem of zero-shot style transfer for text-to-speech (TTS) synthesis of out-of-domain (OOD) custom voice.  To address the problem, the paper proposes GenerSpeech, a generalizable text-to-speech model, which models and controls the style-agnostic (linguistic content) and style-specific (speaker timber, emotion and prosody) speech variations respectively.  More specifically, mix-style layer normalization (MSLN) is proposed to eliminate the style attributes in the linguistic content representation; multi-level style adaptor is adopted for modeling the style representations at global level speaker and emotion characteristics, and the local level (utterance, phoneme and word-level) prosody representations.  Extensive experiments demonstrate the effectiveness of the proposed method in zero-shot style transfer.

**Questions:**

1) It is unclear how the phonetic representation $H_c$ is input to different local style encoders as the query $Q$.  Is $H_c$ phoneme level (before length regulator) or frame level (after length regulator)?  From Fig. 1(a), it seems $H_c$ is phoneme level.  If it is the case, how could the output of Style Adaptor (phoneme level) be added to the output of Content Adaptor (frame level) as the input to Mel Decoder?  The authors need to make this clearer.

2) In Fig. 1(d), it would be better if the authors could explicitly give the additional input of $Q$ (i.e. phonetic representation $H_c$) for Utterance / Phoneme / Word-level local style encoders.

3) In Section 3.3, the paper proposes mix-style layer normalization (MSLN) by considering not only the original style vector $w$ but also the shuffled style vector $ \tilde{w}$.  Experiments also validates the effectiveness of such design.  It would be better the authors give more explanations about MSLN and why it can be adopted for deriving style-agnostic representations.

4) In Section 3.4.1, from Appendix A.2.1, averaging pooling is adopted followed by two sperate full connection (FC) layers to derive the global speaker and emotion information respectively.  It is expected that the averaging pooling of the wav2vec model outputs should carry such speaker and emotion styles.  But it is unclear why wav2vec could be adopted to capture the global style characteristics.  The authors could give more explanations about such design.

5) Although I can guess what does the $S_u$ mean, the authors do not give the definition of $S_u$ in Section 3.4.2.


**Limitations:**

The authors address the impacts and limitations in Appendix G.

**Strengths And Weaknesses:**

Strengths:

1) Towards zero-shot style transfer of OOD voice, the paper proposes a method by decomposing the speech variations into style-agnostic and style-specific parts.  The idea is a kind of speech representation disentanglement which is quite useful for voice cloning and/or style transferring.

2) The paper incorporates several techniques to improve the generalization ability of the proposed method, including mix-style layer normalization (MSLN), multi-level style adaptor, and the flow-based post-net.

3) The samples in the provided demo webpage do show the superiority of the proposed method over the other comparison methods.  For both parallel style transfer and non-parallel transfer, the synthesized results well demonstrate the effectiveness of the proposed method in transferring speaker timbers, emotions, and prosody variations.

Weaknesses:

I don't think this paper has major flaws.  But please refer to further comments for detailed questions.

---

> ### Author Response · Authors · 2022-08-02
> **Response to Reviewer HPwf**
>
> We are grateful for your positive review and valuable feedback, and we hope our response fully resolves your concern.
>
> **[About the phonetic representation $\mathcal{H}_c$.]**
>
> The phonetic representation $\mathcal{H}_c$ has been expanded as frame-level in the content adaptor. We apologize for the mistake in Fig. 1(a), which has been fixed in the new version of the paper.
>
> **[About the marks in Fig. 1(d).]**
>
> Thanks for the reviewer's suggestion, and the additional input (i.e., $Q$) has been explicitly given in the revised version of the paper.
>
> **[About the proposed mix-style layer normalization (MSLN).]**
>
> Thanks for the reviewer's feedback that requests more explanations about our proposed method. Conditional layer normalization has demonstrated its effectiveness in influencing the hidden activation and final prediction. AdaSpeech [1] and Meta-StyleSpeech [2] utilize the speaker embedding as the conditional information, which adaptively scales and shifts the normalized input features to synthesize speech of various voices.
>
>
> For disentangling style information and learning style-agnostic representation, a straightforward solution is to refine the sequence conditioned on the mismatched style information, which could be regarded as injecting noise to confuse the model and prevent it from generating style-consistent representation. Consequently, the model refines the input features regularized by perturbed style and learns generalizable style-invariant content representation. To further ensure diversity and avoid over-fitting, we perturb the style information by randomly mixing the shuffled vectors with a shuffle rate $\lambda$ sampled from the Beta distribution.
>
>
>
> **[About the design of global style encoder.]**
>
> Wav2vec 2.0 is a representation learning framework following a two-stage training process of pre-training and fine-tuning, which has proven its efficiency in learning latent features and even generalizing well to out-of-distribution recordings. Recently, wav2vec 2.0 has been further adopted in various downstream tasks and demonstrated the SOTA results [3]. Inspired by this, we consider the fine-tuned wav2vec 2.0 model as the global style encoder to generate discriminative speaker and emotion embeddings, and perform strong robustness to out-of-distribution custom voices.
>
> **[About the definition of $\mathcal{S}_u$]**
>
> In this work, we use $\mathcal{S}_u$ to denote the phoneme-level style latent representation derived in the style adaptor. Thanks for the reviewer's reminder, and we have attached this definition to the new version of the paper.
>
> **[References]**
>
>
> [1] Chen M, Tan X, Li B, et al. Adaspeech: Adaptive text to speech for custom voice[J]. ICLR, 2021.
>
> [2] Min D, Lee D B, Yang E, et al. Meta-stylespeech: Multi-speaker adaptive text-to-speech generation[C]//International Conference on Machine Learning. PMLR, 2021: 7748-7759.
>
> [3] Wang, Y et al. A fine-tuned wav2vec 2.0/hubert benchmark for speech emotion recognition, speaker verification and spoken language understanding. arXiv preprint arXiv:2111.02735.
>
>
>
> Again, we thank the reviewer for the insightful reviews and "Accept" recommendation for our paper.

---

### Author Response · Authors · 2022-08-02
**Paper Revision**

We thank all reviewers for the constructive feedback. Here we summarize the revision of the manuscript according to the comments and suggestions of reviewers:

- In section 3.3, we include more insights and explanations on designing mix-style layer normalization (MSLN).
- We update Fig. 1 and provide more details.
- In section 3.4.2, we provide the definition of $\mathcal{S}_p$.
- In section 4.3, we bold the definition of abbreviations.
- In section 4.4, we include the detailed fine-tuning setting and put this in Appendix E.
- In Appendix F in the supplementary material, we plot the mel-spectrograms and corresponding pitch tracks generated by different TTS systems for ablation.

---

### Author Response · Authors · 2022-08-07
**Looking forward to further feedback**

Dear Reviewers,

Thank you again for the great efforts and the valuable comments.

We have carefully addressed the main concerns in detail. We hope you might find the response satisfactory. As the end of the rebuttal phase is approaching, we would be grateful if we could hear your feedback regarding our answers to the reviews. We will be very happy to clarify any remaining points (if any).

Thanks in advance,

Paper 4823 authors

---

### Author Response · Authors · 2022-08-09
**Summary of the response**


To all reviewers, ACs, and PCs:

We thank all reviewers for the valuable suggestions with their effort and time. Your comments have improved our work. Here we summarize our effort in addressing reviewers' concerns, and please refer to the responses to each reviewer for more details.

**Extensive experimental comparison to demonstrate the effectiveness of proposed modules.**

- Per the reviewers' suggestions, we have extensively included the expressive Fs2 with post-net for comparison, suggesting the superiority of GenerSpeech in modeling fine-grained style patterns (e.g., local rises and falls of the pitch and stress).

- To prove that global style attributes have been eliminated from the linguistic content representation, we conduct a toy experiment on speaker and emotion classification. The phonetic representation with MLSN has witnessed a distinct decrease in classification accuracy, which verifies the effectiveness of MSLN in disentangling the style information.

- In response to the reviewer's question, we include our preliminary ablation studies for deciding the optimal choice of the VQ size.

**More detailed explanations on model design.**

- As shown in our responses, we discuss the insights of designing MSLN for disentangling style information and learning style-agnostic representation. MSLN refines the sequence conditioned on the mismatched style information, which could be regarded as injecting noise to confuse the model and prevent it from generating style-consistent representation.

- For global (speaker and emotion) style learning, we explain why considering the fine-tuned wav2vec 2.0 model as the style encoder to generate discriminative embeddings.

- For fine-grained style learning, we visualize the mel-spectrograms and corresponding pitch tracks generated by different TTS systems for ablation, presenting the effectiveness of the local style adaptor in modeling multi-level style latent representations.

**More precise definitions and presentation.**

- Per the reviewers' suggestions, we revise Fig. 1(a) to include more annotations for clear definitions.

- As shown in our responses, we present the detailed fine-tuning setting and loss objective for a more precise presentation.

In the meanwhile, we revise the manuscript according to the comments and suggestions of reviewers. We hope all reviewers can carefully read our detailed feedback and re-consider your rating to give a new model a chance. Feel free to ask more questions if you have any time. We are always happy to have a further discussion and answer more questions raised by you.

Best regards, Authors

---

### Meta-Review · Area_Chair_oCJN · 2022-09-09

**Recommendation:** Accept
**Confidence:** Certain

**Metareview:**

All 3 reviewers agree that the paper is novel, technically strong and experimentally convincing. THis paper should be accepted.

**Award:**

No

---

### Decision · Program_Chairs · 2022-09-14

Accept